# Machine Learning Techniques to Enhance Event Reconstruction in Water Cherenkov Detectors †

**Nicholas Prouse *** , **Patrick de Perio** and **Wojciech Fedorko** on behalf of the Hyper-Kamiokande Collaboration

TRIUMF, 4004 Wesbrook Mall, Vancouver, BC V6T 2A3, Canada; pdeperio@triumf.ca (P.d.P.);
wfedorko@triumf.ca (W.F.)
* Correspondence: nprouse@triumf.ca
† Presented at the 23rd International Workshop on Neutrinos from Accelerators, Salt Lake City, UT, USA,
30–31 July 2022.

**Abstract:** Hyper-Kamiokande (Hyper-K) is the next-generation water Cherenkov neutrino experiment, building on the success of its predecessor Super-Kamiokande. To match the increased precision and reduced statistical errors of the new detectors, improvements to event reconstruction and event selection are required to suppress backgrounds and minimise systematic errors. Machine learning has the potential to provide these enhancements, enabling the precision measurements that Hyper-K aims to perform. This paper provides an overview of the areas where machine learning is being explored for Hyper-K's water Cherenkov detectors. Results using various network architectures are presented, along with comparisons to traditional methods and a discussion of the challenges and future plans for applying machine learning techniques.

**Keywords:** neutrino detectors; event reconstruction; machine learning

## 1. Introduction

Hyper-Kamiokande [1] (HK) is the next-generation neutrino experiment in Japan, building on the success of Kamiokande, Super-Kamiokande (SK), and T2K. The physics program includes neutrino oscillation and neutrino astrophysics by observing accelerator, atmospheric, solar, supernova and other astrophysical neutrinos, as well as probes for new physics, including proton decay searches and indirect dark matter searches. The experiment will involve two new water Cherenkov (WC) detectors that are planned to start operation in 2027.

Construction is underway on the far detector (FD), consisting of a 258 kt total (188 kt fiducial) volume of water surrounded by a 20% coverage of 50 cm PMTs with an additional coverage from multi-PMT (mPMT) modules, each consisting of 19 8 cm PMTs. With 8 times the fiducial volume of SK, it will benefit from increased event rates, while the new photosensor technology will provide improved photo-efficiency and timing resolution.

The huge detector target volume and increased neutrino beam power will provide a greatly increased event rate of observed oscillated neutrinos, significantly reducing statistical errors on the measurement of the neutrino oscillation parameters compared to existing neutrino experiments. To achieve the goals of HK, a corresponding reduction is necessary in systematic uncertainties compared to those currently achieved in the T2K experiment. The largest of these uncertainties are in the neutrino cross-sections and interaction models, the neutrino beam flux, and the detector response [2]. A reduction in these uncertainties is being achieved through a combination of advances in analysis techniques, improved detectors, and new measurements of neutrino interactions at the experiment's near and intermediate detectors close to the J-PARC neutrino beam.

The Intermediate Water Cherenkov Detector (IWCD), based on the nuPRISM proposal [3], is planned to be constructed ∼1 km from the beam source to measure the flux and cross-section of neutrinos using the same target and detector technologies as the FD.

The IWCD will consist of a ∼1 kt water volume with ∼500 mPMT modules that provide an improved position, direction, and timing information over 50 cm PMTs, allowing the smaller-sized IWCD to achieve equivalent granularity as the FD. The detector volume of the IWCD will be able to move vertically within a 50 m tall pit, allowing for measurements at different off-axis angles to the beam, providing different neutrino energy spectra.

To exploit new detector technologies and capabilities and achieve the goal of reducing systematic errors, advances in event reconstruction are needed, with challenges posed by the computational requirements of traditional reconstruction and limitations arising from approximations in their physics models [4]. New approaches to event reconstruction are now being explored [5–8], inspired by the success of machine learning techniques in other particle physics and computer vision tasks [9].

## 2. Traditional Event Reconstruction for Water Cherenkov Detectors

Traditional event reconstruction for WC detectors determines the properties and particle types of an observed event by maximising a likelihood or goodness function [10]. This function is based on the probabilities of the observed pattern of hit times and charges at each of the detector's PMTs for a given hypothesis of the particles producing Cherenkov light in the detector. Using this approach and following an algorithm originally developed for MiniBooNE [11], fiTQun is the likelihood-based reconstruction software package used for the reconstruction of high-energy events in HK, T2K, and SK [12].

The single-ring reconstruction of fiTQun maximises the likelihood of the observed hit time and charge at each hit PMT and the likelihood of each unhit PMT, while varying particle position, direction, and energy for each particle type. The ratio of these maximum likelihoods can be used to identify the particle producing the Cherenkov ring.

Multi-ring events are reconstructed by fiTQun using likelihoods calculated assuming contributions from two or more Cherenkov rings. This approach has been successful for neutral pions and other events producing multiple Cherenkov rings, but a dedicated gamma hypothesis in fiTQun, assuming two rings from its conversion to an electron–positron pair, has not been successful. The best performing discriminant for identifying gamma particles is the likelihood ratio of electron and muon hypotheses, where gammas appear less muon-like than electrons due to increased variations in the direction of Cherenkov light from the electromagnetic showers of an electron–positron pair compared to that of a single electron.

Likelihood-based approaches have been successfully used in existing experiments, but their achievable precision is reaching its limit [8]. Adaptations for the smaller IWCD and its geometric complexities require the development of more complex likelihood functions. Improvements require relaxing assumptions used in the likelihoods' construction, resulting in an increased computational complexity. Reconstruction is already the most intensive part of the software chain, for the IWCD single-ring reconstruction with fiTQun takes more than one minute per event, and even longer for the FD with more PMTs. This is beyond what is feasible for some future analyses, so alternative methods are now being explored [13].

## 3. Machine Learning Reconstruction Techniques

Machine learning (ML) has been revolutionary in computer vision and is now becoming common throughout physics applications. ML has the potential to use all information without making model assumptions beyond those of the simulations used to train networks (also used to tune fiTQun). Additionally, once trained, these models use far less computational resources to reconstruct events. The WatChMaL organisation [5] was formed to facilitate the development of machine learning reconstruction for WC detectors, including HK's FD and IWCD. Subsequent studies have reported promising results for identifying neutron captures using various possible network architectures [6] and for particle-type identification (PID) using convolutional neural networks (CNNs) [4] and variational autoencoders [7]. A hybrid approach has also been explored, where the use of ML techniques to generate likelihoods could allow for faster and more accurate results compared to traditional reconstruction methods [8].

Inspired by the revolutionary success of CNNs in image processing [9], and the ability of ResNet to classify broad varieties of images with high accuracy [14], a CNN based on ResNet has been developed to perform classification and regression on events in the IWCD. The input to the network is an image containing both the barrel and end-caps mPMTs of the detector, where each pixel corresponds to a single mPMT with 19 channels for the charge at each of the 19 PMTs. Initial attempts at including an additional 19 channels for the hit times of the PMTs were not successful at improving results; however, further attempts to use this additional information are actively being developed.

The cylindrical geometry of the detector provides two circular end-caps, each containing 88 mPMTs within a $10 \times 10$ grid. The barrel of the cylinder provides a $40 \times 9$ grid of mPMTs. To produce the final $40 \times 38$ pixel image provided to the CNN, first, the barrel is unwrapped and the end-caps are placed above and below; then, each part of the image is duplicated and the layout is reconfigured, as demonstrated in Figure 1. This provides a double-cover of the image that limits adverse effects due to unwrapping the image, and provides physically meaningful circular boundary conditions at both vertical and horizontal boundaries of the image. The network architecture is equivalent to ResNet-18 with its initial $7 \times 7$ pixel convolution layer replaced by a $1 \times 1$ pixel (single mPMT) convolution over the initial 19 channels, corresponding to a convolution of the charges of the 19 PMTs in each mPMT.

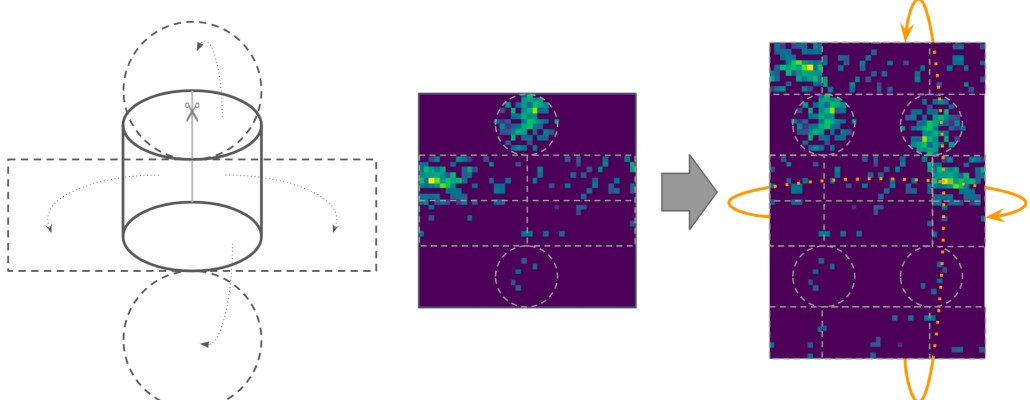

**Figure 1.** Configuration of the cylindrical detector onto a 2D image for CNN. The layout illustration on the left demonstrates how the cylinder is unwrapped onto a 2D surface. Shown on the right, the resulting image is divided into sections outlined by dashed lines, then duplicated and reconfigured into a double cover of the detector surface, with circular boundary conditions indicated by orange arrows.

The use of CNNs requires projecting the detector surface onto a rectangular 2D image of regular pixels. It may be expected that improved results might be achieved by extracting the physical information from the PMT hit times and charges using their positions in 3D space directly. Network architectures acting on point clouds are designed for this task with the PointNet architecture showing the capability of accurately classifying 3D objects using point cloud data [15]. PointNet has been adapted to take points (PMTs) in 3D space, together with features (observations of the PMT) at each point. The input to the network is then each PMT's three spacial position dimensions, hit time, and observed charge. An advantage of this network is that it can be easily adapted to any detector geometry, not requiring that PMTs be positioned in regular grids. For example, this allows a combination of individual 50 cm PMTs and mPMTs, as is for the case for HK FD designs.

## 4. Results

Classification and regression networks based on ResNet and PointNet have been developed for PID and the reconstruction of initial particle position, direction, and energy. A simulated dataset of single-particle events was created using the WCSim software package [16]. The dataset consists of 3 million examples of every electron, muon, and gamma particle, uniformly randomly distributed in the detector volume with isotropic

random directions and uniform random energies of 0 to 1 GeV above Cherenkov's threshold of the particle or its Cherenkov light-producing products. The dataset is partitioned into 1.5 million examples of each particle type for training, 0.3 million for network validation during training, and 1.2 million for evaluating the final model performance. For the more difficult task of separating electrons and gammas, an additional 6 million examples of each were added to the training dataset. Data augmentation is used by including reflections of the detector about each 3D axis, to increase the number of unique examples by a factor of 8.

Figure 2 shows the results for PID, where it is seen that the ML-based approaches are performing well. It should be noted that the traditional fiTQun reconstruction algorithm, designed for the Super-Kamiokande detector, has had its likelihoods re-tuned for the IWCD, but would be expected to perform better if the likelihood construction itself were redesigned to account for complexities of the smaller detector size and mPMT photosensors. Nonetheless, the significant improvement over fiTQun is notable for both the electron vs. muon task and electron vs. gamma task. A very high rejection of muons will be important for the IWCD's large background from the muon neutrino beam, and a statistical separation of single gamma events appears potentially possible in a WC detector for the first time. Of particular note is the reduced performance of fiTQun at low momentum, where events have fewer hits, while ML-based methods appear able to extract all information from these hits, allowing for a good efficiency down to lower energies. For discriminating electrons vs. gammas, ML-based techniques' performance drops near the edges of the energy range used to train the network. Additional data extending the range may improve the results at these edges. It is interesting that while ResNet is outperforming PointNet at electron vs. muon discrimination, the opposite is the case for electron vs. gamma. This is conjectured to be due to the PMT hit timing data, which PointNet has been better able to utilise, helping to extract information from the very start of the electromagnetic shower.

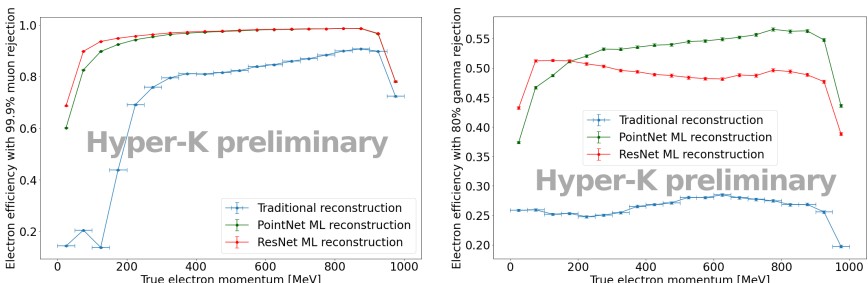

**Figure 2.** Results of PID using ResNet (red), PointNet (green), and fiTQun (blue), showing the electron efficiency when requiring 99.9% muon rejection (**left**) or 80% gamma rejection (**right**) as a function of the electron's true momentum.

Results for the reconstruction of particle position, direction, and energy are shown in Figure 3. In the case of position, PointNet is currently under-performing with a lower resolution than ResNet, and lower than fiTQun when the distance from the detector wall exceeds 125 cm. Improvements over fiTQun are seen when using ResNet for all reconstructed quantities of position, direction, and energy. Energy and direction see general improvements by ResNet over fiTQun. Position reconstruction sees overal improvement, although the position precision of fiTQun meets or exceeds ResNet for distances from the detector wall greater than 220 cm. Of particular note is ResNet's improvement in position reconstruction close to the detector wall. This is expected due to the likelihood functions of fiTQun being originally designed for a larger detector, where approximations in the physical model limit performance close to the detector wall. The strong performance of ML-based methods in the region close to the wall, where fiTQun performance degrades, shows the potential of ML-based reconstruction to reduce systematic uncertainties that could arise from misreconstruction, causing the migration of the position across the boundary of the detector's fiducial volume.

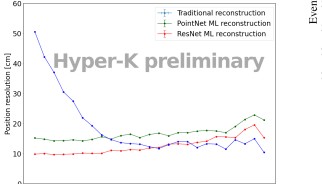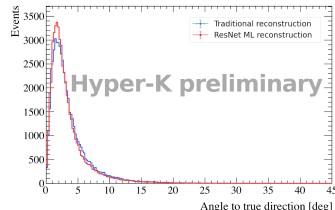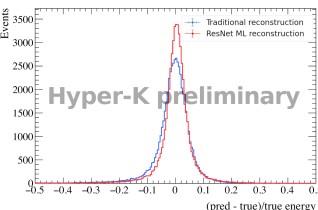

**Figure 3.** Results of reconstruction, showing on the (**left**), the average position resolution as a function of distance from the detector wall for ResNet (red), PointNet (green), and fiTQun (blue); in the (**middle**), the direction resolution; and on the (**right**), the energy resolution.

In addition to improved resolutions and PID capabilities of ML methods, there is also a significant advantage in computational time to perform the reconstruction. Once trained, these networks can process over 100,000 events per minute on one GPU, representing an improvement of five orders of magnitude over traditional reconstruction on CPU, even accounting for processing events through several networks dedicated to different tasks.

Beyond identifying and reconstructing single rings from individual Cherenkov light-producing particles or electromagnetic showers, there is also a need to handle complex events involving multiple particles and Cherenkov rings. One approach to this could be to identify the PMT hit charges in a given event that come from a particular ring, separating from the charge that comes from another ring. Each ring could then be reconstructed individually using the networks trained for single-ring reconstruction. In ML terms, this is a semantic segmentation task, where the image is segmented into pixels belonging to each ring, achieved by designing a network to provide outputs associated with each pixel as opposed to outputs associated with the image as a whole. Two network architectures designed for semantic segmentation based on CNNs, FRRN [17], and U-Net [18] were adapted for IWCD detector data. As a proof-of-concept, these networks were trained to segment the hits coming from two electron-like Cherenkov rings, resulting from the decay of neutral pions. Preliminary results show some promise that the network is able to correctly segment most hits for some events, as seen in the example given in Figure 4; however, further development is needed to extend this approach to the varying numbers and types of particles expected, and to validate the reconstruction of individual rings after segmentation.

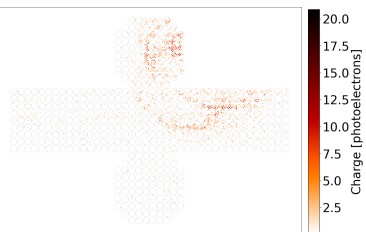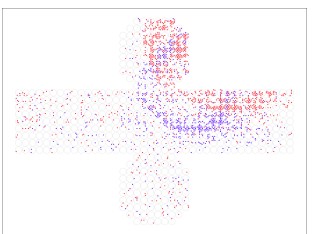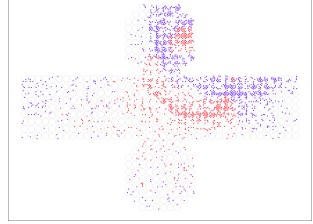

**Figure 4.** Example segmentation of two Cherenkov rings from gammas of a neutral pion decay, showing the simulated charge observed in the detector's PMTs (**left**), true segmentation (**middle**), and reconstructed segmentation using U-Net (**right**). The true and reconstructed segmentation panels display PMT hits from one gamma coloured red and the other coloured blue; the reversal of colour labels identifying the gammas in the reconstructed segmentation occurs by chance due to the network being trained to segment the rings but not to reproduce their arbitrary ordering.

## 5. Conclusions

A new approach for event reconstruction in WC detectors has been presented using deep learning CNN architectures. The results for particle-type identification and reconstruction of particle energy, position, and direction have shown potential to provide improvements over existing likelihood-based reconstruction through a greater precision and reduced computational demands, and the capability to handle multi-ring events by segmentation has also been demonstrated. Such advances will be necessary as part of the

progression to the next generation of neutrino experiments. While the capabilities demonstrated here indicate that the machine learning approach to reconstruction can address the needs of future experiments, further development is still required before these new approaches will be ready to use in a running experiment.

The ML reconstruction performance has only been trained and tested on simulated data. Given the nature of deep learning and concerns about applying a model trained on imperfect simulation, validating these approaches with real data will be essential. Additionally, reconstruction tasks that have been performed so far using ML represent only a subset of those traditionally completed by likelihood methods. To gain the advantage of reduced computational complexity, ML-based methods will need to be extended to fully reconstruct complex multi-ring events with additional particle types. Charged pions have yet to be included, and while segmentation results have shown potential to separate two rings of neutral pions, this provides only one simplified example of a multi-ring event topology and has not yet been integrated into a full reconstruction chain. The upcoming Water Cherenkov Test-beam Experiment [19] will provide an opportunity to address these remaining challenges with data from a WC detector with a beam of tagged particles.

In summary, traditional maximum likelihood event reconstruction is starting to limit the analyses that may be possible with future WC detectors. Significantly reduced reconstruction computation time and potential improvements for reconstruction performance from ML have been demonstrated as a possible route to overcome the limitations of traditional methods. While further development is needed to expand the capabilities and test performance on experimental data, progress is expected to continue towards production-ready ML-based reconstruction for the next generation of WC neutrino detectors.

**Author Contributions:** N.P.: Conceptualization, data curation, formal analysis, investigation, methodology, supervision, software, validation, visualization, writing—original draft. P.d.P.: Conceptualization, funding acquisition, project administration, resources, supervision, writing—review & editing. W.F.: Methodology, resources, supervision, writing—review & editing. All authors have read and agreed to the published version of the manuscript.

**Funding:** This research was funded by the Government of Canada's New Frontiers in Research Fund (NFRF) [NFRFE-2019-00278] and Natural Sciences and Engineering Research Council of Canada (NSERC) [SAPPJ-2019-00036].

**Institutional Review Board Statement:** Not applicable.

**Informed Consent Statement:** Not applicable.

**Data Availability Statement:** The data presented in this study are available on request from the corresponding author. The data are not publicly available due to the large size of the datasets analysed.

**Conflicts of Interest:** The authors declare no conflict of interest. The funders had no role in the design of the study; in the collection, analyses, or interpretation of data; in the writing of the manuscript; or in the decision to publish the results.

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
