# Peer review of "Machine Learning Techniques to Enhance Event Reconstruction in Water Cherenkov Detectors†"

_psf, doi:10.3390/psf8010063_

Round 1
Reviewer 1 Report
The authors offer a comprehensive overview of applying various machine learning techniques to event reconstruction in water Cherenkov detectors, in contrast with traditional methods. The manuscript is generally well-structured, clearly articulated, and effectively describes the methodology, providing a sufficient background. It holds substantial relevance and potential for the Hyper-K experiment. Although the authors have done commendable work in presenting and discussing their research, there is room for enhancing the depth and thoroughness of the discussions to further elevate the manuscript. Adding a "Summary" section summarizing key findings, potential challenges, and future directions would also enhance its clarity and impact.
Specific comments for improvement are:
1. Lines 23-25: Please provide examples to clarify what statistics you're reducing errors in and which systematic uncertainties should be reduced.
2. While detailed references have been incorporated in Sections 2 and 3, it could be advantageous to also cite them in the final paragraph of the Introduction. This would bolster the statements made there and provide immediate access to the source materials for the reader.
3. References supporting statements in the first sentence and final paragraph of Section 2 would enhance credibility.
4. Please elaborate on why a CNN model based on ResNet was chosen over other architectures and define what "successful" entails for the choice of PointNet for 3D.
5. Please offer insights into the observed performance shifts of both ResNet and PointNet, such as implications of the trends and inflection points.
6. A correction is needed at Line 138: It appears that PointNet only underperforms fiTQun when the event distance exceeds roughly 130 cm. Similarly, it seems that ResNet tends to fall short of fiTQun's performance when the event distance is greater than approximately 220 cm.
7. Line 143: Clarify what "this" refers to. Are you suggesting adopting a hybrid method for events occurring outside the detector's fiducial volume?
8. The results for particle direction and energy reconstruction (right two subplots in Figure 3) are not discussed and need to be addressed.
9. Figure 4 requires color labels.
10. The paper could end stronger with a more detailed summary of the current status and future work related to these machine learning architectures. Specifically, how prepared are they for production, and what challenges remain even for single rings?
Author Response
Thank you very much for taking the time to review this manuscript and for the thoughtful and helpful suggestions. I agree with all the comments and have made changes to address them. Please find the detailed responses to all specific comments below, with the corresponding revisions in the re-submitted files and attached pdf.
Specific comments for improvement are:
1. Lines 23-25: Please provide examples to clarify what statistics you're reducing errors in and which systematic uncertainties should be reduced.
Changes have been made to clarify the statistical errors are reduced by the increased neutrino event rate at HK compared to T2K and that the systemactic error improvements in HK address the specific largest uncertainties in T2K's measurements:
"The huge detector target volume and increased neutrino beam power will provide a greatly increased event rate of observed oscillated neutrinos, significantly reducing statistical errors on the measurement of the neutrino oscillation parameters compared to existing neutrino experiments. To achieve the goals of HK, a corresponding reduction is necessary in systematic uncertainties compared to those currently achieved in the T2K experiment. The largest of these uncertainties are in the neutrino cross-sections and interaction models, the neutrino beam flux, and the detector response. Reduction of these uncertainties is being achieved through a combination of advances in analysis techniques, improved detectors and new measurements of neutrino interactions at the experiment's near and intermediate detectors close to the J-PARC neutrino beam."
2. While detailed references have been incorporated in Sections 2 and 3, it could be advantageous to also cite them in the final paragraph of the Introduction. This would bolster the statements made there and provide immediate access to the source materials for the reader.
Additional references have been added to the introduction, particularly in the final paragraph as suggested.
3. References supporting statements in the first sentence and final paragraph of Section 2 would enhance credibility.
Three additional references have been cited for the first sentence and final paragraph of section 2.
4. Please elaborate on why a CNN model based on ResNet was chosen over other architectures and define what "successful" entails for the choice of PointNet for 3D.
Additional clarifications that ResNet was chosen due to "the ability of ResNet to classify broad varieties of images with high accuracy" and PointNet " showing the capability of accurately classifying 3D objects using point cloud data".
5. Please offer insights into the observed performance shifts of both ResNet and PointNet, such as implications of the trends and inflection points.
Additional detail about the shape of the performance curves has been added:
"Of particular note is the reduced performance of fiTQun at low momentum, where events have fewer hits, while ML-based methods appear able to extract all information from these hits allowing good efficiency down to lower energies. For discriminating electrons vs gammas, ML-based techniques performance drops near the edges of the energy range used to train the network. Additional data extending the range may improve the results at these edges."
6. A correction is needed at Line 138: It appears that PointNet only underperforms fiTQun when the event distance exceeds roughly 130 cm. Similarly, it seems that ResNet tends to fall short of fiTQun's performance when the event distance is greater than approximately 220 cm.
Addressed below
7. Line 143: Clarify what "this" refers to. Are you suggesting adopting a hybrid method for events occurring outside the detector's fiducial volume?
Addressed below
8. The results for particle direction and energy reconstruction (right two subplots in Figure 3) are not discussed and need to be addressed.
The above three comments (6, 7, 8) have been addressed through changes to the relevant paragraph:
"Results for reconstruction of particle position, direction and energy are shown in Figure 3. In the case of position, PointNet is currently under-performing with lower resolution than ResNet, and lower than fiTQun when the distance from the detector wall exceeds 125 cm. Improvements over fiTQun are seen when using ResNet for all reconstructed quantities of position, direction and energy. Energy and direction see general improvements by ResNet over fiTQun. Position reconstruction sees overal improvement although the position precision of fiTQun meets or exceeds ResNet for distances from the detector wall greater than 220 cm. Of particular note is ResNet's improvement in position reconstruction close to the detector wall. This is expected due to the likelihood functions of fiTQun being originally designed for a larger detector, where approximations in the physical model limit performance close to the detector wall. The strong performance of ML-based methods in the region close to the wall, where fiTQun performance degrades, shows the potential of ML-based reconstruction to reduce systematic uncertainties that could arise from misreconstruction causing migration of the position across the boundary of the detector's fiducial volume."
9. Figure 4 requires color labels.
A color legend has been added for the left panel and explanation of the colors for the middle and right panels has also been added to the figure caption.
10. The paper could end stronger with a more detailed summary of the current status and future work related to these machine learning architectures. Specifically, how prepared are they for production, and what challenges remain even for single rings?
A conclusion section has been added to provide these details.